# Phenotypic and Histological Distribution Analysis Identify Mast Cell Heterogeneity in Non-Small Cell Lung Cancer

**DOI:** 10.3390/cancers14061394

**Published:** 2022-03-09

**Authors:** Edouard Leveque, Axel Rouch, Charlotte Syrykh, Julien Mazières, Laurent Brouchet, Salvatore Valitutti, Eric Espinosa, Fanny Lafouresse

**Affiliations:** 1Centre de Recherche en Cancérologie de Toulouse (CRCT), UMR1037, INSERM, UMR5071, CNRS, Université Toulouse 3, 31037 Toulouse, France; edouard.leveque@inserm.fr (E.L.); axel.rouch@hotmail.fr (A.R.); salvatore.valitutti@inserm.fr (S.V.); eric.espinosa@inserm.fr (E.E.); 2Thoracic Surgery Department, Hôpital Larrey, CHU Toulouse, 31000 Toulouse, France; brouchet.l@chu-toulouse.fr; 3Department of Pathology, Institut Universitaire du Cancer—Oncopole de Toulouse, 31059 Toulouse, France; syrykh.charlotte@iuct-oncopole.fr; 4Thoracic Oncology Department, Hôpital Larrey, CHU Toulouse, 31000 Toulouse, France; mazieres.j@chu-toulouse.fr

**Keywords:** tumor associated-mast cells, non-small cell lung cancer, integrin alpha-E

## Abstract

**Simple Summary:**

During the fight against tumor, some cells of the immune system such as cytotoxic lymphocytes eliminate tumoral cells while others such as tumor-associated macrophages favor tumor development. Mast cells (MCs) are multifaceted immune cells whose role in cancer is still poorly understood. Moreover, MCs are poorly characterized in the context of cancer and their presence in the tumor microenvironment has been reported to be either associated with good or bad prognosis. In this pilot study we characterized tumor-associated MCs (TAMCs) in lung cancer. We showed that TAMCs exhibited a typical phenotype and can be classified in two subsets according to alphaE integrin (CD103) expression. CD103^+^ TAMCs appeared more mature, more prone to interact with CD4^+^ T cells, and located closer to cancer cells than their CD103^−^ counterpart. This study revealed that a high frequency of total TAMC correlated with better overall survival and progression free survival in patients and underlined MC heterogeneity in cancer.

**Abstract:**

Mast cells (MCs) are multifaceted innate immune cells often present in the tumor microenvironment (TME). However, MCs have been only barely characterized in studies focusing on global immune infiltrate phenotyping. Consequently, their role in cancer is still poorly understood. Furthermore, their prognosis value is confusing since MCs have been associated with good and bad (or both) prognosis depending on the cancer type. In this pilot study performed on a surgical cohort of 48 patients with Non-Small Cell Lung Cancer (NSCLC), we characterized MC population within the TME and in matching non-lesional lung areas, by multicolor flow cytometry and confocal microscopy. Our results showed that tumor-associated MCs (TAMCs) harbor a distinct phenotype as compared with MCs present in non-lesional counterpart of the lung. Moreover, we found two TAMCs subsets based on the expression of CD103 (also named alphaE integrin). CD103^+^ TAMCs appeared more mature, more prone to interact with CD4^+^ T cells, and located closer to cancer cells than their CD103^−^ counterpart. In spite of these characteristics, we did not observe a prognosis advantage of a high frequency of CD103^+^ TAMCs, while a high frequency of total TAMC correlated with better overall survival and progression free survival. Together, this study reveals that TAMCs constitute a heterogeneous population and indicates that MC subsets should be considered for patients’ stratification and management in future research.

## 1. Introduction

Non-small cell lung cancer (NSCLC) counts for about 85% of lung cancers and remains the leading cause of cancer-related death worldwide with only 17% of patients alive 5 years after diagnosis [1,2,3]. Lung adenocarcinoma (LUAD, approximately 40–50% of cases) and lung squamous cell carcinoma (LUSC, approximately 20–30% of cases) are the predominant histological subtypes of NSCLC [4]. During tumor development, the tumor cells provoke notorious changes in the tissue environment by reprogramming tissue physiology to their own benefit. Resident immune cells respond to these changes in tissue homeostasis in a still poorly understood manner. Nevertheless, the immune system plays a crucial role in the fight against cancer as highlighted by the benefits of immunotherapies targeting immune checkpoints on NSCLC patients’ survival [5,6].

In the tumor microenvironment (TME), immune cells playing key roles in the cancer-immune cycle have been mostly investigated, including lymphocytes but also to a lesser extent innate immune populations such as dendritic cells and macrophages [7,8,9]. Mast cells (MCs) are innate immune cells that are particularly abundant at barrier sites. MC can be divided in two subsets according to their location [10]: connective tissue-associated MCs are constitutive and tissue-resident MCs present in skin and serosa while mucosal MCs are present in mucosa and notably in lungs and enriched following immune challenge. In human lungs, mucosal MCs reside in bronchi, bronchioles, and alveola, while connective tissue-associated MCs are found in pulmonary vessels and pleura [11]. MCs arise from mast cell progenitors originating from the bone marrow that infiltrate tissues thanks to Vascular Cell Adhesion Molecule-1 (VCAM-1) expressed by endothelial cells and tissue chemotactic factors such as Stem Cell Factor (SCF). On their side, MC progenitors express α4β7 integrins allowing them to bind VCAM-1 and infiltrate lungs where they terminate their differentiation [12,13]. MCs are able to release a large array of mediators in response to modifications of their environment and are well known to participate to inflammation and allergy but also to tissue repair and homeostasis. MCs were often observed in TME and their ability to produce pro and anti-inflammatory mediators render it difficult to hypothesize a beneficial or a detrimental role in cancer. Indeed, mast cells were recurrently suspected to play either pro- or anti-tumoral roles [14,15,16]. While in some cancers such as gastric, bladder or thyroid cancer, MCs were always associated with poor prognosis, in others cancers the role played by mast cells is still elusive (reviewed in [16,17]). In NSCLC, MC role is controversial [18,19,20,21]. Independently of their prognostic impact, MCs are endowed with several immune functions that can influence the TME. Therefore, their potential role in cancer is of high interest as highlighted by an increasing number of reviews discussing the topic [14,15,16]. However, while MCs have been occasionally quantified in the TME, they remain poorly defined and characterized. Whether the MC phenotype is altered by the TME and leads to heterogeneity similarly to what was reported for other myeloid cells such as macrophages, monocytes, dendritic cells, or neutrophils [22] remains to be elucidated. In this study, we explored MC phenotype, location, and prognostic value in NSCLC. We described TAMC heterogeneity and identified two TAMC subpopulations according to their CD103 (integrin alpha-E) expression.

## 2. Material and Methods

### 2.1. Patients Samples

The NSCLCs were obtained from lung resection surgery in the thoracic surgery department of the Toulouse University Hospital. Patients were included prospectively from 1 August 2018 to 31 August 2021. Patient samples were obtained after informed consent in accordance with the declaration of Helsinki. Collection was authorized by the Ministry of Higher Education and Research (DC-2021-4673) and a transfer agreement (contract 20200820019) was obtained after approbation by ethical committees. The inclusion date corresponds to the date of the lung cancer surgery. The average follow-up was 20.61 months (1.70–43.17). Patients’ characteristics are described in Table 1.

Kaplan–Meier curves of the OS et PFS were performed depending on the percentage of TAMC in the tumor immune infiltrate and the frequency of CD103positive TAMC. Patients were assigned to the TAMC^high^ subgroup when TAMC frequency was higher than 1.43% (median of MC frequency) and to the CD103^high^TAMC when the frequency of CD103positive TAMC was higher than 10.95% (median of CD103^+^ MC frequency).

### 2.2. Histology

Freshly resected lung tissue samples were fixed in 4% paraformaldehyde over-night at 4 °C. Following fixation, samples were washed in PBS and incubated over-night in sucrose 30%. Samples were then embedded in O.C.T compounds (Agar Scientific, Stansted, UK) and stored at −80 °C until cryo-sectioning with a cryostat Microtom HM 550 (Thermo Scientific, Waltham, MA, USA). Six micrometers of tissue sections were incubated in citrate buffer (pH 6.0, Sigma Aldrich, Saint Louis, MO, USA); heated to 95 °C and saturated with a blocking buffer (PBS with 3% BSA, 10% FCS, 10% HS, 0.1% saponin) for 30 min at room temperature; and then stained with primary antibodies in PBS 3% BSA, 0.1% saponin at 4 °C overnight. Sections were next washed four times in PBS 3% BSA, 0.1% saponin and incubated with secondary antibodies diluted in PBS 3% BSA, 0.1% saponin for 2 h at room temperature. Sections were washed three times, counterstained with DAPI (4′,6-Diamidino-2-Phenylindole, Dihydrochloride, 1 μg/mL, ThermoFisher, Waltham, MA, USA) for 5 min to label cell nuclei and washed two times in PBS 3% BSA, 0.1% saponin, two times in PBS 3% BSA, and two times in PBS before mounting them in 90% glycerol-PBS containing 2.5% DABCO (Sigma). Primary and secondary antibodies used are detailed below in the key resource table. Primary antibodies used to stain CD103 and cytokeratin were both from rabbit. For double labelling sections with the same host species primary antibodies, anti-CD103 staining revealed by its secondary antibody and anti-pan-cytokeratin staining revealed by its secondary antibody were performed successively and goat Fab fragments (Clinisciences, Nanterre, France) anti-rabbit IgG, secondary antibody, were used to block rabbit IgG surface between the two stainings for 1 h at room temperature. Images were acquired using a LSM 780 confocal microscope (Zeiss, Oberkochen, Germany) with a ×63 plan-Apochromat objective (1.4 oil) and a 0.6 numerical zoom, using tile scan mode. Images acquired were analyzed using Zen (Carl Zeiss Microscopy) and IMARIS 9.8 (Oxford instrument, Abingdon, UK) software. Using Imaris software, cells were segmented with surface and spots modules and the shortest distances between segmented elements were quantified. Patient’s tissues were randomly selected and tissue sections showing a high tissue autofluorescence, poor cryopreservation, or cut-damaged were discarded from the analysis.

### 2.3. Flow Cytometry

Freshly resected lung tissue samples were cut into small pieces and digested for 1 h at 37 °C using the tumor dissociation kit (Milteniy Biotech, Bergisch Gladbach, Germany) and the gentleMACS Dissociator (Milteniy Biotech). Digested samples were filtered through a 100 µm cell strainer, washed, and red blood cell lysis (Gibco, Waltham, MA, USA) was performed. CD45^+^ immune cells were positively sorted using CD45 (TIL) microbeads (Milteniy Biotech) according to manufacturer recommendations. Recovered cells were stained for 30 min with fluorochrome-labelled primary antibodies in PBS 1% HS, 1% FCS at 4 °C. Cell viability was ascertained by labeling with fixable viability dyes (eBioscience, San Diego, CA, USA). For intracellular staining, cells were washed in PBS and next fixed in PFA 4% for 10 min at room temperature and permeabilized with PBS 1%, FCS 1%, HS 0.1% saponin (permeabilization buffer) for 10 min. Cells were next incubated with the following antibodies VEGF-PE (R&D systems) and OPN-FITC (R&D systems, Minneapolis, MI, USA) in permeabilization buffer for 45 min at RT. Cells were next washed in PBS and proceeded to flow cytometry analysis. Stained cells were acquired using a Fortessa flow cytometer (BD Biosciences, Franklin Lakes, NJ, USA) and analyzed with FlowJo V10.4.2 software (TreeStar, Antwerp, Belgium).

### 2.4. Statistical Analysis

Statistical tests were performed with Graph Pad Prism V9 software (GraphPad Software, Inc., San Diego, CA, USA). Tests performed are indicated in the figure legends. Non parametric tests (Wilcoxon signed paired rank) were performed when group distribution failed normality or variance homogeneity. All *p* values are two-sided, (* *p* < 0.05; ** *p* < 0.01; *** *p* < 0.001; **** *p* < 0.0001 and ns, not significant). 

### 2.5. Key Resources Table

Please see key resources in Table 2.

## 3. Results

### 3.1. NSCLC Patient Cohort

During the study period, tumoral and non-lesional samples from 48 patients who benefited from major lung resection for NSCLC in the thoracic surgery department of the Toulouse University Hospital were analyzed. The mean age was 64.02 and the sex ratio was 1.4 men per women. The major part (91.67%) of the patients had a past history of smoking. In this cohort, histological diagnosis was LUAD for 72.92% of patients, LUSC for 18.75% of patients, and large cell carcinoma and pleomorphic carcinoma for 8.33% of patients. Pathological Tumor Node Metastasis (pTNM) staging of this cohort of NSCLC was mostly stage III (56.25%). All patient’s characteristics are shown in Table 1.

### 3.2. Analysis of Mast Cells in the TME

While MCs play critical roles in lung homeostasis and are associated with several lung pathologies [23], these cells have been poorly characterized in NSCLC. We first analyzed MCs by immuno-histology in lung tumors and matched non-lesional lung tissues sampled at distance from the tumor mass in the resected lobe, from seven randomly selected patients in the cohort. Tissues were stained for tryptase (MCs) and CD45 (immune cells) and large tile scans were imaged by confocal microscopy (Figure 1A). The number of MCs (CD45^+^ Tryptase^+^ cells) per mm^2^ was quantified using Imaris software (Figure 1B,C). Mast cells were observed in both tumor and non-lesional lung tissues, showing an important variability between patients. MC number was slightly, but not significantly, lower in the tumor microenvironment (TME) than in non-lesional lung tissue. To get a better insight into MC quantification in tissues, we next analyzed mast cell frequency by flow cytometry in our entire cohort of patients. The immune infiltrate was isolated from single-cell suspension of digested tissues, based on CD45 cell sorting, and MCs were next identified as CD45^+^ CD14^−^ CD117^+^ (c-kit) FcεRI^+^ cells by multiparametric flow cytometry (Figure 1D). MCs were identified in both tumoral and non-lesional tissues of all NSCLC patients (46 patients, 2 missing data). MC frequency among CD45^+^ cells showed a high variability between patients and was slightly reduced (but not significantly, *p* = 0.057) in the TME (1.79% ± 1.56, Mean ± SD) as compared to non-lesional tissues (3.04% ± 3.73 Mean ± SD) (Figure 1E).

To get insights into tumor-associated MC (TAMC) location in relation to tumoral cells, we next analyzed tryptase^+^ cells and cytokeratin^+^ cell (tumoral cells) distribution pattern in the TME by confocal microscopy (Figure 2A). We first noticed that TAMCs globally followed the same distribution pattern as the CD45^+^ immune infiltrate (Figure 2A,B). To analyze cell positioning in detail, we defined three location types for a cell, according to its location relative to tumor cell rich regions: (1) inside, (2) at the edge, or (3) outside the tumor cell-rich regions (Figure 2C). Imaris software allowed to classify mast cells (Figure 2D) and CD45^+^ cells (Figure 2E) according to these location types. This analysis revealed that MCs mainly resided outside tumor cell-rich regions.

Because MCs are usually found in the vicinity of blood vessels, we analyzed the TAMC location relative to blood vessels. We found that most TAMCs were located at less than 100 µm of CD31^+^ endothelial cells, suggesting that TAMCs keep this strategic positioning in the TME (Figure 2F,G).

### 3.3. Mast Cell Phenotype Is Impacted by the Tumor Microenvironment

To further characterize TAMCs in NSCLC patients, we compared their phenotype to patient-matched non-lesional MCs by flow cytometry. In tumors, we found that TAMCs exhibited a higher granularity (SSC-A parameter) than their non-tumoral counterparts (Figure 3A) and that they expressed higher levels of FcεRI (Figure 3B) and lower levels of CD117 (Figure 3C). According to previous reports showing that MC precursors harbor a SSC^low^ CD117^high^ FcεRI^low^ phenotype [24,25], the observed TAMC phenotype suggests that TAMC exhibited a more mature phenotype than their non-lesional counterparts. To go further, we analyzed β7 integrin expression (involved in MC precursors homing to mucosa and progressively lost across MC maturation [24]) and CD103, the epithelial-binding αE integrin [26]. β7 integrin expression was decreased while CD103 was increased in TAMCs (Figure 3D,E). Moreover, this analysis revealed the existence of two TAMC populations according to CD103 expression. Because high β7 integrin expression was related in mouse models to inducible mucosal mast cells [26], we analyzed whether TAMC were related to mucosal or connective tissue MC. MRGPRX2 (a receptor restricted to connective tissue MC [27,28]) was not expressed in TAMCs or MCs in non-lesional areas (Figure 3F) indicating that MCs mostly pertained to the mucosal type. Moreover, we observed an increased expression of CD38 in TAMC (Figure 3G). CD38 expression by MC was previously reported [29] but appeared variable according to the tissue and pathophysiology [30]. Nevertheless, CD38 increased expression in TAMCs might be associated to specific MC functions in the TME.

Because TAMCs exhibited some features reminiscent to mature mast cells, we next analyzed the expression of molecules in relation to MC functions. Regarding MC ability to interact with CD4^+^ T cells, MHC class II and CD80 costimulatory molecules were expressed by about 50% and 40% of MC, respectively, independently of their lesional or non-lesional location (Figure 4A,B) with a slight tendency to lower MHC class II molecules for TAMC. Usually, MCs do not express MHC class II molecules at steady state. This result indicates that the tumor impacted not only TAMCs but also MC residing in the non-lesional part of the lungs. The InterCellular Adhesion Molecule-1 (ICAM-1; an adhesion molecule important for immune synapse formation with other immune cells) expression level was similar in TAMCs as compared to non-lesional MCs (Figure 4C). Finally, we analyzed Vascular Endothelial Growth Factor (VEGF) and osteopontin expression (two critical molecules in cancer biology) [31] and showed that a higher proportion of TAMCs express VEGF and osteopontin (Figure 4D,E).

Collectively these results indicate that TAMCs exhibit a mature phenotype associated with specific biological functions such as antigen presentation and detrimental cancer-related mediators.

### 3.4. CD103 Identifies Two Distinct Mast Cell Populations in the TME

Since we identified two distinct populations of TAMCs based on their expression of CD103 (Figure 3D), we next investigated the phenotype of CD103^−^ and CD103^+^ TAMC subsets by flow cytometry. CD103^+^ TAMCs were more granular (Figure 5A), expressed a lower level of CD117 (Figure 5B) and a comparable level of FcεRI (Figure 5C) than CD103^−^ TAMCs. In addition, CD103^+^ TAMCs expressed a higher level of MHC-class II molecules (HLA-DP-DQ-DR) (Figure 5D), CD80 (Figure 5E), and ICAM-1 (Figure 5F), suggesting a more mature phenotype and a higher capacity for antigen presentation. Figure 5G shows an example of CD103^+^, MHC-II^+^ TAMC. Furthermore, while CD103^+^ TAMCs expressed a lower level of osteopontin (Figure 5H), their expression level of VEGF was similar to their CD103^−^ TAMC counterparts (Figure 5I). Confocal microscopy analysis showed that CD103^+^ TAMCs were located closer to tumoral cells than their CD103^−^ counterparts (Figure 5J,K).

Taken together these results identified two distinct populations of TAMCs: CD103^+^ TAMCs harboring an increased capacity for antigen presentation and located closer to cancer cells than CD103^−^ TAMCs.

### 3.5. Mast Cells Are Associated with Better Survival in NSCLC

To determine the contribution of TAMC on the NSCLC prognosis in our prospective cohort, patients were separated in TAMC^high^ and TAMC^low^ relatively to the median of MC frequency in TME quantified by flow cytometry (Figure 1E; median 1.43%). The main clinical characteristics of TAMC^high^ and TAMC^low^ are depicted in Table 3 and show no statistical differences between the two groups.

Figure 6A,B show respectively that patients in the TAMC^high^ group had a significantly better overall survival and progression-free survival, than TAMC^low^ patients. Next, to evaluate the impact of MC phenotypic heterogeneity in our surgical cohort, we assessed the prognosis influence of CD103 expression in TAMC quantified for 38 patients by flow cytometry (Figure 3E). Patients were divided into two groups according to the median of TAMC frequency expressing CD103 (median 10.95%), and Kaplan–Meier curves showed no significant difference for the overall survival and the progression-free survival between CD103^high^ and CD103^low^ (Appendix A). The main clinical characteristics are shown in Appendix A and revealed a statistically significant reduced percentage of LUAD in the TAMC-CD103^high^ group (47.37% versus 89.47%) and corresponding increased percentage of SCC (36.84% versus 5.26%) compared to TAMC-CD103^low^ group. This difference is further highlighted in Appendix A, showing that SCC patients expressed a higher percentage of CD103^+^TAMC than LUAD patients. Together these results show that TAMC are associated with good prognosis in NSCLC patients while TAMC-CD103^high^ are enriched in SCC patients and did not confer any advantages for the patient’s survival in our surgical cohort.

## 4. Discussion

In this study, we characterized the phenotype of human TAMCs in lung cancer tissue samples and evaluated phenotypic and functional differences with their non-lesional counterparts. On the basis of previous studies carried out in mouse [24,32], the TAMCs we described here exhibited a more mature phenotype (CD117^low^ FcεRI^high^ SSC^high^ and integrin β7^low^) than MCs located in non-lesional lung. This observation, together with a slightly reduced presence of MCs in tumor, might be interpreted as a reduced turnover of MC in tumors.

We observed in TAMCs some phenotypic differences as compared to normal human lung MCs such as CD80 and CD103 expression [29]. Surprisingly we found that a substantial part of TAMCs express CD103. CD103 (αE)β7 integrin binds to E-cadherin expressed by epithelial cells. The CD103 role is well known to allow CD103^+^ tissue-resident memory T cells to adhere to epithelial cells and reside in tissues [33]. In cancer, CD8^+^CD103^+^ TIL participate in tumor cell killing and cytokine production within the TME [34]. CD103 has been shown to promote CTL lytic granules polarization toward tumoral target cells by sustaining immune synapse formation [35]. CD103 expression is poorly documented in MCs, notably in human MCs. In a mouse model, CD103 was found to be induced in MC by TGF-β1 [36]. Moreover, TGF-β1 might help MC progenitors that infiltrate the TME to maturate toward CD103^+^ mucosal MCs, as shown in maturating mouse MCs [36]. These observations are compatible with CD103 induction in MCs by a TGFβ1-rich TME. While the role of CD103 in MCs is still elusive, it is conceivable that CD103 might favor MCs adhesion to epithelial cells. Indeed, intraepithelial MCs induced by T. spiralis were reported positive for CD103 [37]. Furthermore, and in favor of this hypothesis, we observed that CD103^+^ MCs located closer to tumoral cells than their CD103^−^ counterparts, suggesting that CD103 allows MC to adhere to epithelial cells. Whether CD103 signaling in synergy with key well-characterized MC receptors drives some MC response (as it was reported for CD8^+^ T cells [35]) remains to be investigated.

We observed that an important part of both lesional or non-lesional MCs exhibited an antigen-presenting cell (APC) phenotype (MHC class-II and CD80 costimulatory molecules expression). We and others have previously reported that MCs can serve as APC upon priming with IFN-γ [38,39]. The fact that MC harbor APC phenotype in non-lesional lung suggests that either tumor presence induces perturbations in the whole organ or IFN-γ produced inside the TME diffuses outside the tumoral mass. Nevertheless, the MHC-class II molecule expression level was slightly lower in the TME, suggesting that suppressive mechanisms operate in the TME. The CD103^+^MC subset exhibited a stronger expression of ICAM-1, CD80, and MHC-class II molecules, indicating that they are more prone to antigen presentation than their CD103^−^ counterparts. This observation suggests that CD103^+^MC found in closer contact with tumoral epithelial cells in the TME, might present tumor-associated antigen to local CD4^+^ T cells. The efficacy of CD103^+^ MCs to serve as APC for CD4^+^ T cells and the outcome of this cooperation for CD4^+^ T cells in term of differentiation toward fully differentiated helper T cells or toward anergic helper T cells or even regulatory T cells, remains to be elucidated.

Our results also show that the frequency of MC expressing osteopontin and VEGF was augmented in the TME. Osteopontin, beyond its role in biomineralization and bone remodeling, binds to integrins or CD44 (expressed by immunocytes) and participates in leukocytes recruitment in tissues [40,41]. Osteopontin was found up-regulated in NSCLC, participating in the crosstalk between tumoral cells and the host microenvironment and favoring tumor progression and immune evasion [42,43]. Nevertheless, acknowledging the fact that several leukocytes and tumoral cell produce osteopontin, the real impact of MC-produced osteopontin remains elusive. VEGF production by MC, as a potential detrimental factors in cancer, has been extensively discussed [14,44]. Indeed, beside macrophages, MCs are an important source of VEGF and might therefore participate in neoangiogenesis. We observed that CD103^+^MCs expressed VEGF at a similar level to their CD103^−^ counterparts while osteopontin expression was reduced. According to the potential detrimental role of these two molecules, CD103^+^ MC appears more favorable than CD103^−^MC. This point is substantiated by the fact that they appear more mature and with increased APC capabilities, suggesting a better aptitude to promote inflammation and efficient CD4^+^ T cell responses. In line with this hypothesis, patient’s TME rich in MCs were associated with better survival in our patient cohort. This result is in agreement with some previous studies [19,45]. It is worthy to note that MC tryptase might also have a beneficial impact as it has been reported previously in melanoma [46,47]. However, and surprisingly, no clear advantages were shown in our study for patients exhibiting strong CD103^+^ MC profile. This study is limited by the size of the cohort, and the potential different impacts of CD103^+^ and CD103^−^ MC on cancer progression need to be investigated in larger patient cohorts in the future. Furthermore, it is important to take into consideration that a higher frequency of CD103^+^MC was found in SCC patients, highlighting the needs to perform additional studies with a larger SCC patient cohort in the future.

## 5. Conclusions

This analysis reveals the presence of distinct MC subpopulations (on the basis of CD103 expression) in the TME with potentially distinct functions. This heterogeneity is reminiscent to macrophage or dendritic cell diversity and needs to be further explored notably to know whether MCs are phenotypically and functionally molded by the growing tumor. This study shows that patients exhibiting higher frequencies of MC in the TME have better overall survival and progression-free survival.

## Figures and Tables

**Figure 1 cancers-14-01394-f001:**
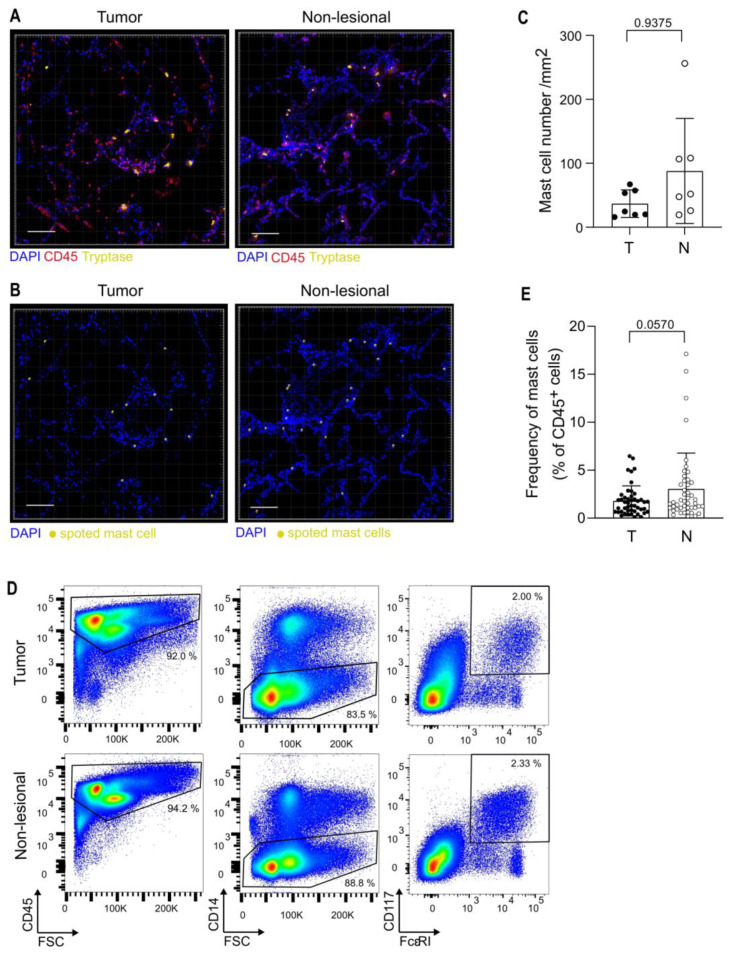
Mast cells are relatively abundant within the tumor microenvironment of NSCLC patients. (**A**) Representative confocal images of tumoral and non-lesional lung tissue sections stained for Tryptase (Yellow), CD45 (red), and DAPI (blue). Scale bar, 100 µm. (**B**) Tryptase^+^ mast cells (yellow) were identified and quantified using Imaris software. Each dot represents one cell. Scale bar, 100 µm. (**C**) MC number in tumoral (T) and non-lesional (N) matched tissues (*n* = 7 NSCLC patients). (**D**) MC gating strategy. Representative dot-plots showing CD117^+^ FcεRI^+^ MCs gated on CD45^+^ CD14^−^ cells isolated from tumoral and non-lesional lung tissues. (**E**) Frequency of CD117^+^ FcεRI^+^ MCs among single, live, CD45^+^ cells in tumoral (T) and non-lesional (N) lung tissues in 46 NSCLC patients. (**C**,**E**) Mean and SD are shown, Wilcoxon paired t-test, *p*-values are indicated.

**Figure 2 cancers-14-01394-f002:**
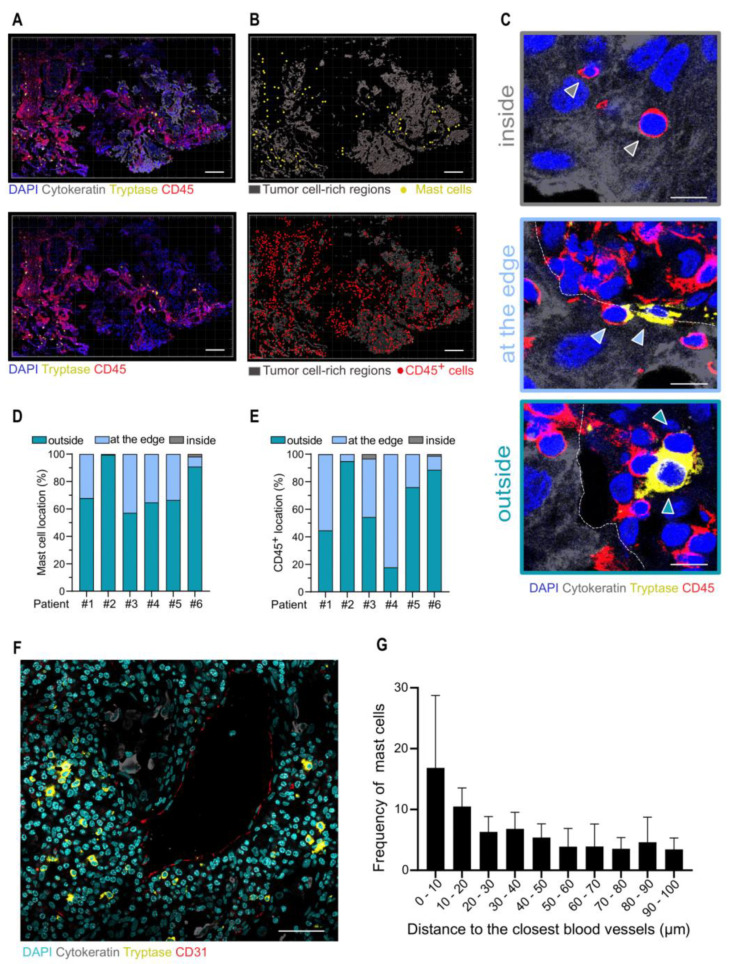
Mast cell cartography in NSCLC tumors. (**A**) Representative confocal large tile scan images of tumoral lung tissue section stained for Cytokeratin (grey), Tryptase (Yellow), and CD45 (red); Scale bar, 200 µm. (**B**) Tryptase+ MCs (yellow) and CD45+ cells (red) were identified using Imaris software; Scale, 200 µm. (**C**) Representative examples showing immune cells inside cytokeratin-rich regions (top), at the edge of the cytokeratin-rich regions (middle) or outside cytokeratin-rich regions (bottom). Arrows indicate examples of immune cells classified in the presented categories. Dashed lines indicate the cytokeratin-rich region edge. Scale bar, 10µm. (**D**,**E**) Frequency of MCs (**D**) and CD45+ cells (**E**) in the indicated categories (*n* = 6 NSCLC patients). (**F**) Representative image showing MCs (Tryptase+, yellow), endothelial cells (CD31+, red), cancer cells (pan-cytokeratin+, grey), and nucleus (DAPI, cyan) in the TME. Scale bar = 50 µm. (**G**) MC distribution according to their shortest distance to blood vessels. Results are from four patients. Bars represent mean and SD.

**Figure 3 cancers-14-01394-f003:**
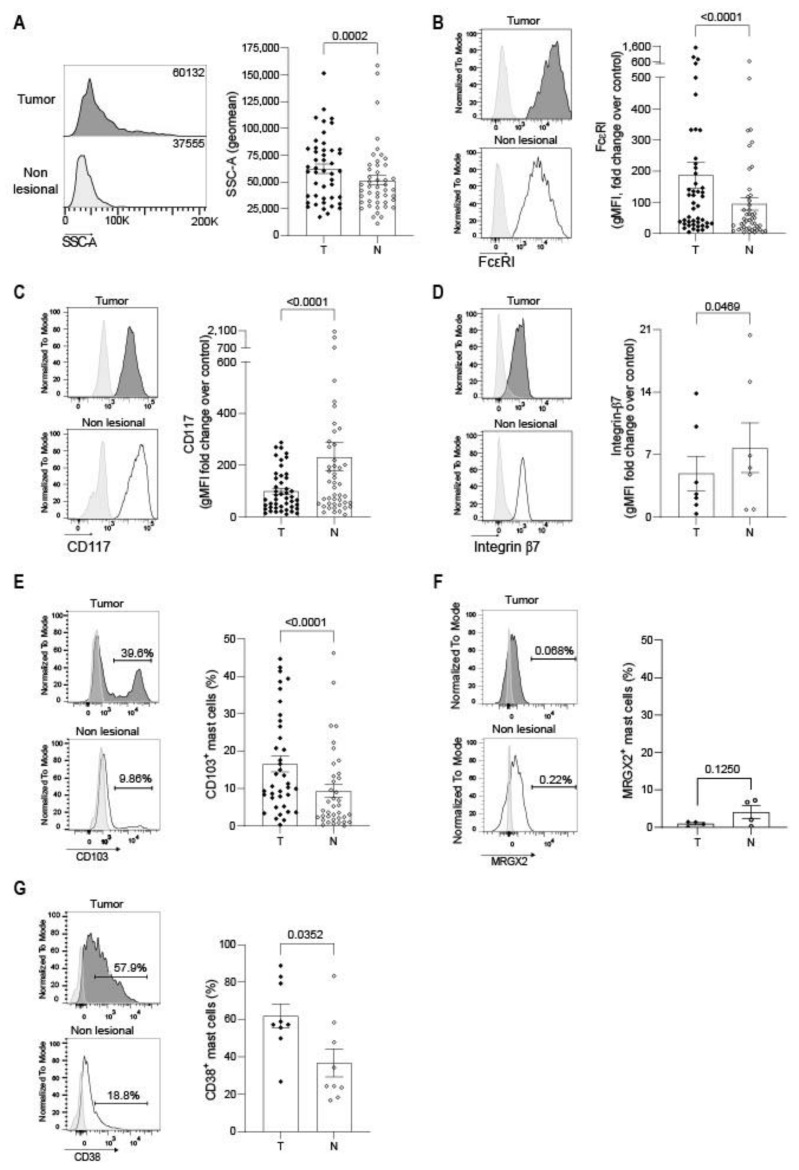
Mast cell phenotype is impacted by the tumor microenvironment. (**A**–**G**) Left panels: representative histograms showing the granularity (SSC intensity) (**A**) or the expression of the indicated marker (**B**–**G**) in MCs from tumoral and non-lesional-matched lung tissues analyzed by flow cytometry. Right panels: data are presented as gMFI fold change over unstained control (**B**–**D**) or cell frequency (**E**–**G**) for the indicated marker (mean and SEM). Each dot represents one patient. Wilcoxon signed-rank test.

**Figure 4 cancers-14-01394-f004:**
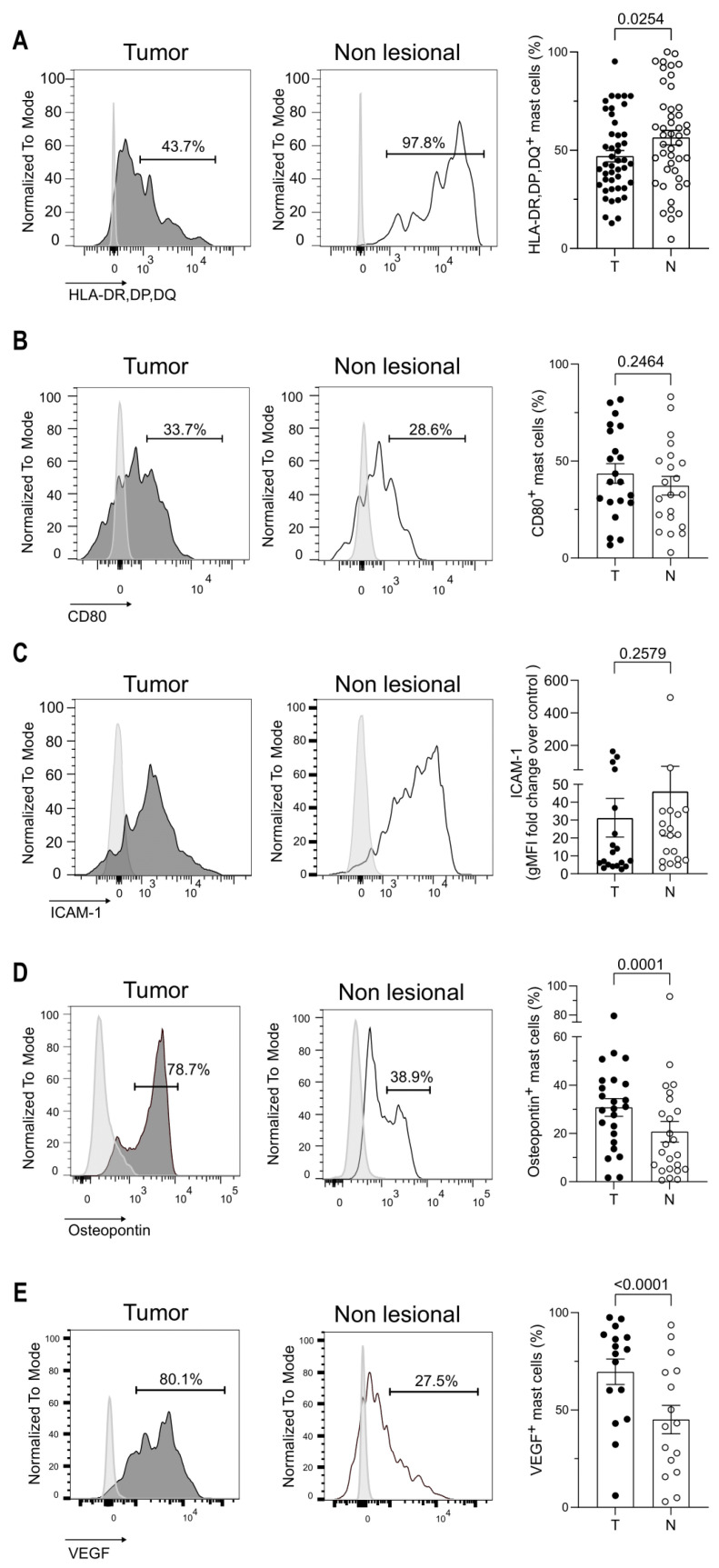
Key markers of MC-associated functions are altered in the TME. (**A**–**E**). Left panels: Representative histograms showing the expression of the indicated marker in MCs from tumoral and non-lesional-matched lung tissues analyzed by flow cytometry. Right panels: data are presented as cell frequency (**A**,**C**–**E**) or gMFI fold change over unstained control (**B**) for the indicated marker (mean and SEM). Each dot represents one patient. Wilcoxon signed-rank test.

**Figure 5 cancers-14-01394-f005:**
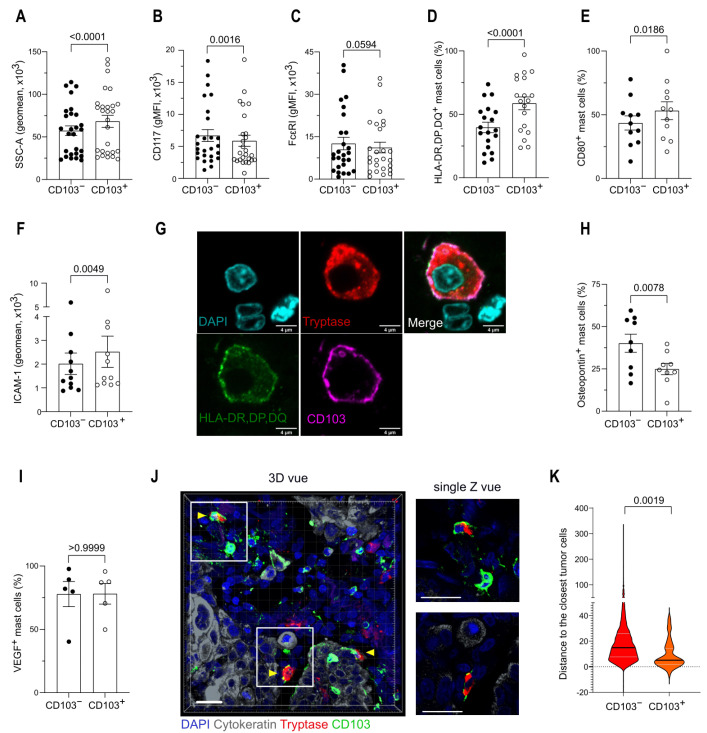
CD103 expression defines two distinct TAMC populations. (**A**–**C**) Granularity (SSC), CD117, and FceRI expression (gMFI) in CD103^−^ and CD103^+^ TAMCs, analyzed by flow cytometry. (**D**,**E**) Frequency of HLA-DR, DP, DQ, and CD80-positive mast cells for CD103^−^ and CD103^+^ TAMCs, analyzed by flow cytometry. (**F**) ICAM-1 expression (gMFI) in CD103^−^ and CD103^+^ TAMCs, analyzed by flow cytometry. (**G**) Representative confocal image showing a TAMC stained for DAPI (light blue), Tryptase (red), CD103 (violet), and HLA-DP-DQ-DR molecules (green). Scale bar, 4 µm. (**H**,**I**) Frequency of osteopontin and VEGF-positive mast cells for CD103^−^ and CD103^+^ TAMCs, analyzed by flow cytometry. (**A**–**F**,**H**,**I**) Each dot represents one patient; mean and SEM are shown. Wilcoxon signed-rank test. (**J**) Representative 3D-reconstruction and single Z-stack magnified image of CD103^+^ TAMC; DAPI (blue), Cytokeratin (gray), Tryptase (red), and CD103 (green). Scale bar, 20 µm. (**K**) Shortest distance between CD103^−^ or CD103^+^ TAMC and cytokeratin-rich region (pooled data from 5 NSCLC patients). Unpaired t-test. Data are presented as violin plot with median (black) and quartiles (white).

**Figure 6 cancers-14-01394-f006:**
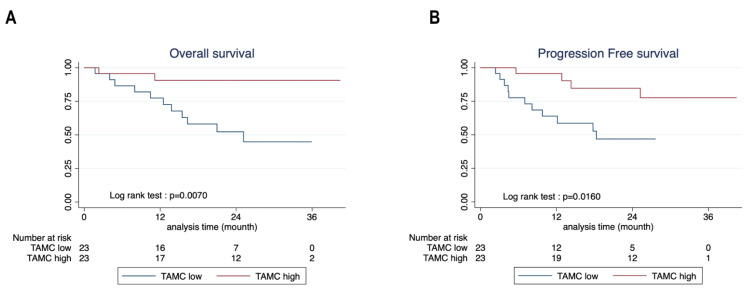
High TAMC frequency is associated with a good prognosis in the surgical cohort. (**A**,**B**) Kaplan–Meier curves showing the (**A**) overall survival (OS) and (**B**) progression-free survival (PFS) of 23 TAMC high (red) and 23 TAMC low (blue) patients with surgical stage NSCLC. TAMC high and TAMC low were separated according to the median of TAMC frequency in the immune infiltrate as measured by flow cytometry (Figure 1E).

**Table 1 cancers-14-01394-t001:** Patient’s characteristics.

Characteristics	Patients *n* = 48
Sex	
Male	28 (58.33)
Female	20 (41.67)
Age, year	64.02 (61.39–66.65)
Smoking history	44 (91.67)
Histology	
LUAD	35 (72.92)
LUSC	9 (18.75)
Other	4 (8.33)
pTNM staging	
I	9 (18.75)
II	12 (25.00)
III	27 (56.25)
IV	0

**Table 2 cancers-14-01394-t002:** Key Resources Table.

Reagent or Resource	Source	Identifier	Dilution
Antibodies—Flow Cytometry			
Mouse Anti-Human CD117 (c-Kit) BUV395	BD Biosciences	Cat# 745733 RRID: AB_2743207	1/50
Mouse anti-Human FcεRI APC	eBiosciences	Cat# 17-5899-42 RRID: AB_10671394	1/50
Mouse anti-Human HLA-DR,DP,DQ BV650	BD Biosciences	Cat# 740582 RRID:AB_2740041	1/100
Mouse anti-Human CD54 BV786	BD Biosciences	Cat# 740978 RRID: AB_2740602	1/50
Mouse anti-Human CD80 PE-Cy5	BD Biosciences	Cat# 559370 RRID: AB_397239	1/25
Mouse anti-Human CD38 Pe-vio 770	Miltenyi Biotec	Cat# 130-118-982 RRID: AB_2751601	1/50
Mouse anti-Human CD45 Amcyan	Miltenyi Biotec	Cat# 130-110-638 RRID: AB_2658245	1/100
Mouse anti-Human CD103 BV 421	BD Biosciences	Cat# 563882 RRID: AB_2738464	1/50
Mouse anti-Human CD14 PECF-594	BD Biosciences	Cat# 562335 RRID: AB_11153663	1/100
Mouse anti-Human MRGX2 PE	Biolegend	Cat# 359003 RRID: AB_2562300	1/25
Rat anti-Integrin b7 BV650	BD Biosciences	Cat# 564284 RRID:AB_2738729	1/50
Mouse anti-Human osteopontin FITC	R and D systems	Cat# IC14331F RRID: AB_10888852	1/25
Mouse anti-Human VEGF PE	R and D systems	Cat# IC2931P RRID: AB_357311	1/50
Antibodies—Histology			
Mouse (IgG1) anti Human-Tryptase	Agilent	Cat# M7052 RRID: AB_2206478	1/200
Rabbit anti-Human CD103	Abcam	Cat# ab224202 RRID: AB_2891141	1/100
Rabbit anti-Human CD103	Diagomics	Cat# BSB2864	1/100
Rabbit anti-Human pan-cytokeratin	Thermo Fisher Scientific	Cat# PA127114 RRID: AB_780038	1/100
Mouse (IgG2a) anti-Human CD31	Thermo Fisher Scientific	Cat# MA3100 RRID: AB_223516	1/100
Rat anti-Human CD45	Thermo Fisher Scientific	Cat# MA517687 RRID: AB_2539077	1/100
Goat anti-Rabbit IgG (H + L) Highly Cross-Adsorbed Secondary Antibody, Alexa Fluor Plus 555	Thermo Fisher Scientific	Cat# A32732 RRID: AB_2633281	1/200
Goat anti-Rabbit IgG (H + L) Highly Cross-Adsorbed Secondary Antibody, Alexa Fluor Plus 647	Thermo Fisher Scientific	Cat# A32733 RRID: AB_2633282	1/200
Goat anti-Rabbit IgG (H + L) Highly Cross-Adsorbed Secondary Antibody, Alexa Fluor 660	Thermo Fisher Scientific	Cat# A-21074 RRID: AB_2535735	1/200
Goat anti-mouse IgG2a Cross-Adsorbed Secondary Antibody, Alexa Fluor 594	Thermo Fisher Scientific	Cat# A21135 RRID: AB_1500827	1/200
Goat anti-mouse IgG1 Cross-Adsorbed Secondary Antibody, Alexa Fluor 488	Thermo Fisher Scientific	Cat# A21121 RRID: AB_2535764	1/200
Goat anti-Rat IgG (H + L) Highly Cross-Adsorbed Secondary Antibody, Alexa Fluor 546	Thermo Fisher Scientific	Cat# A-11081 RRID: AB_141738	1/200
Chemicals, Peptides, and Recombinant Proteins			
eBioscience™ Fixable Viability Dye eFluor™ 780	Thermo Fisher Scientific	65-0865-18	1/500
Citrate Buffer, pH 6.0, 10×, Antigen Retriever	Sigma-Aldrich	Cat# C9999	
ACK lysing buffer	Gibco	Cat# A1049201	
DAPI	Thermofisher	Cat# D1306	1 μg/mL
Critical Commercial Assays			
Tumor Dissociation Kit, human	Miltenyi Biotec	Cat# 130-095-929	
Software and Algorithms			
GraphPad Prism 9	graphpad.com	N/A	
Flowjo 10.0	Flowjo.com		
Zen	Carl Zeiss Microscopy		
ImageJ	ImageJ		
Imaris	Imaris		

**Table 3 cancers-14-01394-t003:** TAMC low and TAMC high patient’s characteristics.

Characteristics	TAMC Low *n* = 23 (%)	TAMC High *n* = 23 (%)	*p* Value
Sex			0.765
Male	14 (60.87)	13 (56.52)	
Female	9 (39.13)	10 (43.48)	
Age, year	65.39 (62.48–68.29)	62.52 (57.75–67.29)	0.292
Smoking history	22 (95.65)	20 (86.96)	0.295
Histology			
LUAD	19 (82.61)	14 (60.87)	0.102
LUSC	3 (13.04)	6 (26.09)	0.265
Other	1 (4.35)	3 (13.04)	0.129
pTNM staging			
I	5 (21.74)	4 (17.39)	0.710
II	3 (13.04)	8 (34.78)	0.223
III	15 (65.22)	11 (47.83)	0.429
IV	0 (0)	0 (0)	

## Data Availability

The datasets used and analyzed during the study are available from the corresponding author on reasonable request.

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
