# Peer review of "Phenotypic and Histological Distribution Analysis Identify Mast Cell Heterogeneity in Non-Small Cell Lung Cancer"

_cancers, 2022, doi:10.3390/cancers14061394_

Round 1

Reviewer 1 Report

The article "PHENOTYPIC AND HISTOLOGICAL DISTRIBUTION ANALYSIS IDENTIFY MAST CELL HETEROGENEITY IN NON-SMALL CELL LUNG CANCER" is devoted to topical issues of the biology of tumor-associated mast cells in non-small cell lung cancer. The features of CD103+ MC shown by the authors, despite the absence of significant prognostic advantages, are useful in further studies of the pro- and antitumorogenic effects of MC secretome components. At the same time, the reviewer had some questions, the answers to which can improve the quality and significance of the scientific data provided.

  1. According to some research groups, the increase in the population of MC in the tumor is a positive prognosis. This was also shown in this study. However, for the detection of MC, the authors used only one specific protease, tryptase. How objective is it to evaluate the entire population of mast cells in an organ based only on tryptase detection? In the opinion of the reviewer and based on literature data (Atiakshin et al., 2021), the profile of specific proteases, regardless of belonging to the mucosal or connective tissue subpopulations, is quite variable. Mast cells with the Tryptase- Chymase+ CPA3+ phenotype, as well as tryptase-chymase+CPA3-, can constitute a significant pool in the total number of MCs, depending on adaptive or pathological processes. Works on the biology of specific MC proteases are well known. If the authors did not identify other specific proteases (chymase and CPA3), then the discussion should be focused on the fact that we are talking only about the features of tryptase-positive mast cells.
  2. How can the authors explain the significant difference in the results of CD103 expression in lung mast cells compared to the data of the study by Rönnberg E et al, (2022, see list of recommended sources) presented in Figure 2, in which the authors detected Cd103 in only 1% of MC?. How stable was the intensity of CD103 expression in CD103+ MC?
  3. When discussing the results obtained by the authors, it were useful to use information about the functional potential of tryptase, which affects, among other things, the proliferative activity of tumor cells (Pejler G et al., 2021, Grujic M et al., 2020, Rabelo Melo F et al., 2019 ).
  4. Line 55 – Please clarify information about integrins: "On their side, MC progenitors express 4 7 integrins allowing ....".
  5. Line 83 - Where is Table number 1? It is absent in the materials available to the reviewer.
  6. Line 86: Why did the authors choose these values - 1.43 and 10.95???
  7. In the "Key resources table", it is necessary to distinguish - which antibodies were used for IHC and which for Flow Cytometry, and indicate the antibody dilution used.
  8. Line 142 – Please support with citations the sentence: "While MCs play critical roles in lung homeostasis and are associated with several lung pathologies, these cells have been poorly characterized in NSCLC".
  9. Need to improve the quality of the images in Figure 2A. Also, it is necessary to use general notations so that the text might be easier for the reader to understand. If the authors use the designations DAPI, Cytokeratin, Tryptase, CD45 in one figure, the same mode should be preserved in other figures. When improving the quality of micrographs, the authors need to make mast cells visible in Figure 2B, in this version they are not visualized. Generally, the authors might think about the presentations of images in a better quality in their future publications.
  10. Line 199 - How was the MC granularity evaluated? In Fig. 3A, in what units is the SSC-A parameter presented? Was it possible to assess the maturity of MC granules as described in Atiakshin et al., 2021?
  11. How the authors might explain the formation of a more mature MC phenotype in a tumor?
  12. Line 212 - Are the authors aware of the results of studies, in which CD38 was detected not only in the mucous MCs (Atiakshin, 2021)? Here, the point of view on the classification of CD38+ mast cells is somewhat subjective. Did the authors evaluate the intensity of CD38 expression using immunohistochemical analysis of paraffin lung sections?
  13. Line 278. Table 2 is missing from the article.

Рекомендуемый список литературы

  • Atiakshin D, Buchwalow I, Horny P, Tiemann M. Protease profile of normal and neoplastic mast cells in the human bone marrow with special emphasis on systemic mastocytosis. Histochem Cell Biol. 2021 May;155(5):561-580. doi: 10.1007/s00418-021-01964-3. Epub 2021 Jan 25. PMID: 33492488; PMCID: PMC8134284.
  • Rönnberg E, Boey DZH, Ravindran A, Säfholm J, Orre AC, Al-Ameri M, Adner M, Dahlén SE, Dahlin JS, Nilsson G. Immunoprofiling Reveals Novel Mast Cell Receptors and the Continuous Nature of Human Lung Mast Cell Heterogeneity. Front Immunol. 2022 Jan 4;12:804812. doi: 10.3389/fimmu.2021.804812. PMID: 35058936; PMCID: PMC8764255.
  • Pejler G, Alanazi S, Grujic M, Adler J, Olsson AK, Sommerhoff CP, Rabelo Melo F. Mast Cell Tryptase Potentiates Neutrophil Extracellular Trap Formation. J Innate Immun. 2021 Dec 22:1-14. doi: 10.1159/000520972. Epub ahead of print. PMID: 34937018.
  • Grujic M, Hellman L, Gustafson AM, Akula S, Melo FR, Pejler G. Protective role of mouse mast cell tryptase Mcpt6 in melanoma. Pigment Cell Melanoma Res. 2020 Jul;33(4):579-590. doi: 10.1111/pcmr.12859. Epub 2020 Jan 19. PMID: 31894627; PMCID: PMC7317424.
  • Rabelo Melo F, Santosh Martin S, Sommerhoff CP, Pejler G. Exosome-mediated uptake of mast cell tryptase into the nucleus of melanoma cells: a novel axis for regulating tumor cell proliferation and gene expression. Cell Death Dis. 2019 Sep 10;10(9):659. doi: 10.1038/s41419-019-1879-4. PMID: 31506436; PMCID: PMC6736983.
  • Atiakshin D, Samoilova V, Buchwalow I, Tiemann M. Expression of CD38 in Mast Cells: Cytological and Histotopographic Features. Cells. 2021 Sep 22;10(10):2511. doi: 10.3390/cells10102511. PMID: 34685490; PMCID: PMC8534017.

Reviewer 2 Report

The manuscript is well written and presented. The authors identify two mast cell population and that TMACs correlates with a better survival rate in Non-Small Cell Lung Cancer.

Detail statical analysis in the MM section.

How is the MC in a healthy individual? Please include a few words in the discussion section. 

In figure 2, how were analyzed patients selected?

Include study limitation statement in the discussion. 

The conclusion is too general. Add, for instance, the impact of TMACs o survival rate.

MCs are the source of pro-tumorigenic factors and antitumorigenic molecules. Did the authors measure TNF and VEGF from TAMCs or MC subsets in the tumoral area? These results will provide a significant step to increase manuscript quality.

Check for minor typos errors such as line 55.

In line 158, "... was slightly reduced in the TME..." however, there is no statistical difference.
